# The Evolution of Insulation Performance of Fiber-Reinforced Silica Aerogel after High-Temperature Treatment

**DOI:** 10.3390/ma16134888

**Published:** 2023-07-07

**Authors:** Rui Gao, Zhangjian Zhou, Hongbo Zhang, Xiaoge Zhang, Yuming Wu

**Affiliations:** 1School of Materials Science and Engineering, University of Science and Technology Beijing, Beijing 100083, China; gao_rui95@163.com (R.G.);; 2Zhongfa Innovation (Beijing) Energy Conservation Technology Co., Ltd., Beijing 100086, China

**Keywords:** fiber-reinforced silica aerogel, glass wool, heat treatment, thermal conductivity

## Abstract

Fiber-reinforced silica aerogel blankets (FRABs) are an important high-temperature thermal insulation material for industry applications that have emerged in recent years. In order to better understand the performance evolution of FRABs at high temperatures, the effect of heat treatment at different temperatures on the performance of FRABs as well as their base material, hydrophobic silica aerogel powder and glass wool, was investigated. The property evolution of the hydrophobic silica aerogel powder showed two stages with an increase in thermal treatment temperatures. The skeleton structure of the aerogel remained unchanged, but the residual organic chemicals, such as hydrophobic groups, were decomposed when the heat treatment temperature was lower than 400 °C. Above 400 °C, the skeleton began to shrink with the increase in temperature, which led to an increase in thermal conductivity. The structure and room-temperature thermal conductivity of the glass wool blanket were less affected by a heat treatment temperature under 600 °C. Therefore, the performance degradation of FRABs at high temperatures is mainly due to the change in the aerogel powder. The insulation performance of the glass wool and FRAB at high temperatures was studied using a heating table which was designed to simulate working conditions. The energy savings of using FRABs instead of glass fiber were calculated and are discussed here.

## 1. Introduction

The energy crisis and carbon emissions have increasingly become the focus of international attention. One of the most important ways to achieve the goal of energy saving and emission reduction is to use efficient insulation materials in industry and architecture [1].

Hydrophobic silica aerogel has excellent properties such as extremely low density, very low thermal conductivity, and excellent waterproof performance [2,3]. Using hydrophobic silica aerogel as an insulation material to replace traditional insulation materials in the fields of building, high-temperature pipelines, and space exploration can greatly improve energy efficiency and therefore reduce carbon emissions [4,5,6]. 

The current commercially available hydrophobic silica aerogels are quite brittle; therefore, it is usually necessary to composite aerogel powder with other matrix materials (typically glass wool) for industrial applications as insulation materials. A number of aerogel-based composites have been developed and investigated, such as glass fiber–aerogel composites [7,8], expanded perlite–silica aerogel composites [9], polyurethane–silica aerogel composites [10], polypropylene–silica aerogel composites [11], etc. Among them, glass-fiber-reinforced aerogel blankets (FRABs) have been applied in industry [12,13]. For instance, FRABs have been applied to building and shown excellent characteristics, including very low thermal conductivity, a light weight, very good sound isolation, and excellent waterproof performance [14,15,16]. Furthermore, FRABs also have great application prospects in high-temperature environments, such as steam pipes in power plants [17,18] and petrochemical plants [19]. The typical operation temperature is usually between 200 °C and 600 °C. By using FRABs instead of traditional insulation materials (basalt fiber and glass fiber), it is expected that a significant amount of energy can be saved [20]. The investigation of the effect of high temperature on the microstructure stability and insulation performance of FRABs is crucial for engineering applications [21]. 

Many authors have studied the effect of heat treatment temperature on the microstructure changes of laboratory-made aerogels. Pradip B. Sarawade et al. [22] studied the changes in the microstructure of aerogels after heat treatment at temperatures up to 500 °C and found that the hydrophobicity of the silica aerogel spheres gradually decreased with the increase in temperature. After heat treatment at 500 °C, the aerogel microspheres became completely hydrophilic. As the temperature increased, the pore size of the aerogel microspheres increased. Gaosheng Wei et al. [23] measured the thermal conductivity of aerogels at different temperatures using the transient thermal strip method and found that the thermal conductivity increased significantly with an increase in temperature. The aerogel particles were partially sintered, and the pores were destroyed when the temperature rose to 950 °C, resulting in an increase in thermal conductivity and bulk density [24,25]. Huafei Cai et al. [26] investigated the thermal stability of silica aerogel using TEM with an in situ heating device. They found that the silica aerogel first shrank at the initial time (within 2 h) at a temperature between 600 °C and 1000 °C, and then the structure became stable during longer-term heating. Previous studies have shown that the pore structure of aerogel can be destroyed by high temperatures, especially when the temperature is higher than 600 °C, which will lead to a drastic decrease in the insulation performance. Moreover, when the heat treatment temperature is lower than 600 °C, the pore structure of the aerogel does not change significantly over a short period of time (less than 10 h). Nevertheless, the performance change due to structural changes in aerogel particles caused by the high temperatures is not representative of the overall FRAB performance change. 

Although the effect of glass fiber content in fiber-reinforced aerogel composites (FRABs) on their properties has been widely studied [27,28], the current research has little involvement in the relationship between the service temperature and the insulation performance of FRABs. Most of the obtained results were based on calculations and simulations. For example, the total effective thermal conductivity of aerogels and their composites considering the combined effect of conduction and radiation was predicted [29,30]. Changes in the structure and properties of aerogel composites at different temperatures have also been investigated via calculations [31,32]. Miros et al. compared the thermal conductivity of FRABs and mineral wool at different temperatures using the guarded hot plate method [33]. Fang et al. used a model to predict the thermal conductivity of aerogel composites and then verified the accuracy of the model by measuring the high-temperature thermal conductivity using the hot disk method [34]. Furthermore, in recent studies, efforts were mainly focused on the development of new grades of aerogel materials which are still a long way from commercial application. For pure aerogel material, only a glass window combining silica aerogel has been reported as an engineering application [35,36]. FRABs have been widely used in high-temperature industries, but their performance in use has rarely been evaluated, and the results of pure aerogel obtained in the laboratory cannot be used to accurately evaluate the service behavior of composite materials such as FRABs. In order to better understand the service behavior of FRABs, it is necessary to compare and evaluate aerogel, the matrix of glass wool, and FRABs under the same simulated actual operating conditions.

In this study, a novel heating table was designed and constructed to simulate actual operating conditions; therefore, the heat loss of the heat-treated FRAB and glass wool with different thicknesses at different temperatures was able to be measured and compared. These data, combined with the evolution of the microstructure and thermal conductivity of aerogel and FRABs after heat treatment, can help better guide the application of aerogel-based insulation materials. 

## 2. Experimental Materials and Methods

### 2.1. Experimental Material

FRABs and their constituent material glass wool and silica aerogel were provided by JINNA TECH. The silica aerogel was prepared using water glass as the precursor. The wet gel was obtained through a sol-gel route, followed by solvent exchange with ethanol, surface modification with trimethylchlorosilane, and supercritical carbon dioxide drying. The specific surface area of the aerogel is 339.064 m²/g, and the average pore size is 12.69 nm. The glass wool matrix used to make FRABs has a density of 160 kg/m^3^. FRABs were prepared by infiltrating wet gel into glass wool and then supercritical drying. The density of the FRABs is 200 kg/m³, of which the aerogel content is 20 wt.%. 

### 2.2. Heat Treatment Method

In order to explore the change in the thermal conductivity of aerogels after thermal treatment at different temperatures, FRABs and their constituent material glass wool and silica aerogel were heat-treated in a box-type resistance furnace (SX2-12-12, Shanghai Zhetai Machinery Manufacturing Co., Ltd., Shanghai, China) with a heating rate of 3 °C/min. When the temperature reached the specified value (200 °C, 300 °C, 400 °C, 500 °C, 600 °C, and 700 °C), the furnace was maintained at that temperature for 2 h. Then, these samples were naturally cooled to room temperature.

### 2.3. Characterization Methods

#### 2.3.1. Characterization of Apparent Properties

The bulk density of the aerogel samples was calculated by measuring their mass-to-volume ratio. The hydrophobic properties were tested using the following methods: Aerogel powders before and after heat treatment were spread on a polyethylene plate, respectively, as polyethylene does not absorb water. Then, water was dropped on it to observe the droplet spreading characteristics. 

According to the ‘‘Determination of thermal conductivity of non-metallic solid materials—hot wire method”, the TC-3000E thermal conductivity tester (Xiaxi Technology, Xi’an, China) was used to determine the room temperature thermal conductivity of the samples.

The thermal stability of the aerogel samples was analyzed by TG-DTA using an SDT Q600 (TA Instruments, New Castle, DE, USA). The samples were heat-treated under an air atmosphere from room temperature (~25 °C) to 1000 °C with a controlled heating rate of 10 °C/min.

#### 2.3.2. Microstructural Characterization 

In order to study the microstructure of the samples before and after heat treatment, a scanning electron microscope (SEM) (SIGMA 300, Carl Zeiss AG, Oberkochen, Germany) was used. Aerogels and FRABs were fixed on the sample holder using a carbon pad and subsequently coated with platinum for SEM analysis. The surface chemical modification of the aerogels was studied using Fourier transform infrared spectroscopy (FTIR) (TENSOR 37, Bruker, Karlsruhe, Germany). FTIR data for heat-treated (200 °C, 400 °C, 600 °C) aerogel powders were collected.

#### 2.3.3. High Temperature Insulation Performance Test

In order to explore the high-temperature insulation performance of FRABs, we designed the following heating table, as shown in Figure 1. It contains two parts: the console and the heating device. The temperature of the heating element is adjusted by the control instrument; we also call it the working temperature. The heater power and, consequently, the heating pace are adjusted using the power knob. The temperature of the thermocouple is displayed by the display instrument. The testing process is divided into the following three steps: First, put the insulation material evenly into the heating device. Ensuring that there are no gaps between each contact surface is the key to a successful test. Second, turn on the power and set the temperature and then wait for a while until the heating power stabilizes. Third, record data obtained from the thermocouples and heating power.

The temperature on the surface and the heating power of the insulation with different thicknesses and different working temperatures were tested at the same ambient temperature. The temperature on the surface is the temperature of the surface of the tested material. The heating power is the power consumption of the heating device during the test.

## 3. Results and Discussion 

### 3.1. The Effect of Heat Treatment on Silica Aerogel

TG-DTA was used to explore the weight change and potential reaction of aerogel at high temperatures. The result is shown in Figure 2.

There is a strong endothermic peak before 100 °C with a weight loss of 4%. According to the preparation process of the aerogel, it should be caused by the evaporation of residual low-boiling organics inside the aerogel [37]. When the temperature was 400 °C~600 °C, a more obvious weight loss of 8% could be found, which can be judged due to the decomposition of hydrophobic groups in the hydrophobic aerogel at high temperatures [38]. This inference was further supported by Figure 3, which shows how heat treatment affected the hydrophobic properties of the aerogel. When the aerogel was heated at 200 °C, the hydrophobicity of the aerogel remained relatively good; however, when the temperature increased to 400 °C, the hydrophobicity completely disappeared. After 600 °C, the weight loss decreased obviously because the methyl group decomposition had been completed.

Figure 4 shows the infrared spectroscopy results to further confirm the reason for hydrophobicity disappearance. The vibrations at 3666 cm^−1^ and 1650 cm^−1^ were due to residual –OH groups or adsorbed water. The background peak was caused by potassium bromide introduced into the test. The absorption peaks near 1085, 810, and 455 cm^−1^ represented the asymmetric stretching vibration, symmetric stretching vibration, and bending vibration of Si-O-Si [39]. The absorption peaks at 850 cm^−1^ and 2968 cm^−1^ could be attributed to S-C bending vibration and −CH_3_ stretching vibration [40,41]. For the samples heat treated at 200 °C, pronounced absorption peaks were observed in the spectra. In contrast, almost no absorption peaks appeared at these two points in the spectra of the samples heat treated at 400 °C and 600 °C, respectively. There was additional absorption at 3200 cm^−1^ formed by −OH for the 200 °C-treated sample, but it was absent for samples treated at higher temperatures.

The above analysis indicated that when heat treated above 400 °C, the methyl group is pyrolyzed, therefore destroying the hydrophobicity of aerogel. 

The macroscopic morphology of aerogel after heat treatment at different temperatures is shown in Figure 5. It can be clearly seen that the aerogel shrank slightly after heat treatment at 400 °C, while significant volume shrinkage occurred when treated at temperatures higher than 600 °C.

Figure 6 shows the changes in the bulk density and RT thermal conductivity of silica aerogels after heat treatment at different temperatures. When the heat treatment temperature was lower than 300 °C, the bulk density of the aerogel decreased first and then remained stable. The decrease in the bulk density of the aerogel in this case is due to the decomposition of residual organic compounds. Therefore, it can be inferred that the skeleton structure of the aerogel will be stable at temperatures below 300 °C. After heating at 400 °C, the aerogel volume shrank slightly, the weight decreased due to the loss of hydrophobic groups, and the bulk density increased slightly. Then, the bulk density increased obviously along with the increase in heat treatment temperature, especially when the temperature was higher than 600 °C, as shown in Figure 5. Especially, when the temperature reached 900 °C, a severe shrinkage of the aerogel could be found. This indicates that the aerogel skeleton is severely damaged and the pores collapse when heat treated at high temperatures.

The change in thermal conductivity with heat treatment temperature was consistent with the bulk density change. When the heat treatment temperature was lower than 400 °C, the thermal conductivity even decreased slightly due to the volatilization of the organic chemicals. Then, the thermal conductivity increased along with the increase in heat treatment temperature due to the volume shrinkage, especially when the heat treatment temperature was higher than 600 °C.

Figure 7 shows the microstructure of aerogel powders after heat treatment at different temperatures. Untreated silica aerogel showed a highly porous and homogeneous nanostructure, which was similar to that of the samples heat treated at 400 °C. After heat treatment at 600 °C, a small part of the microscopic pore structure of the aerogel was destroyed. Particle diameter and pore sizes changed slightly, and the collapse of the framework was not obvious. For the samples heat-treated at 900 °C, the porous structure collapsed badly, and the particles aggregated and grew up.

According to the above test results, the change in silica aerogel after heat treatment can be divided into two stages. When the heat treatment temperature was lower than 400 °C in the first stage, the residual organic compound or hydrophobic groups introduced during the preparation process were decomposed, while the skeleton structure of the aerogel remained unchanged, resulting in a slight reduction in thermal conductivity. The second stage occurred when the heat treatment temperature was higher than 400 °C, which resulted in volume shrinkage and the destruction of the aerogel skeleton, therefore increasing the thermal conductivity. This phenomenon was more significant at temperatures above 600 °C.

### 3.2. The Effect of Heat Treatment on Glass Wool

The glass wool matrix was also heat-treated, and the weight loss and thermal conductivity were measured, as shown in Figure 8. When the heat treatment temperature exceeded 300 °C, the weight of the glass wool decreased by 2% due to the decomposition of residual organic additives added during the preparation of glass wool. According to Figure 8, room temperature thermal conductivity did not change significantly after heat treatment from 200 °C to 600 °C. Glass fibers had low solid-state thermal conductivity due to their low density and thin glass fibers. Meanwhile, fine glass fibers can capture and fix the air between the fibers, thereby preventing heat transmission by convection and limiting gaseous heat conduction by minimizing collisions between gas molecules. Therefore, the RT thermal conductivity of glass wool is very low and will not change with the heat treatment temperature.

### 3.3. The Effect of Heat Treatment on FRAB

The relationship between the weight and thermal conductivity of FRABs after heat treatment from RT to 600 °C is shown in Figure 9. The mass loss under 400 °C was considered to be the decomposition of organic groups (−CH_3_ groups) in the composites. This could be confirmed by the droplet-spreading characteristics of aerogel after heat treatment as shown in Figure 10. In Figure 10b, no droplets could be seen on the surface of the FRABs because droplets penetrate quickly into the materials. The FRABs experience a sharp weight loss from 400 °C to 600 °C, which is consistent with Liao’s research [42]. The RT thermal conductivity of FRAB decreased first and then increased with the increase in heat treatment temperature, which had the same trend as aerogel shown in Figure 11. The structure and thermal conductivity of the glass wool were less affected by the heat treatment temperature under 600 °C in Section 3.2. Therefore, the performance change of FRABs at high temperatures was mainly due to the change in the aerogel. 

The microstructure of the FRAB samples was characterized through SEM images, as shown in Figure 12. At low magnification in Figure 12a,d,g, fibers were easily observed because of the high contrast. The fiber structure did not change with the increase in the heat treatment temperature. The change in the aerogel microstructure in the FRABs along with the heat treatment temperature is shown in Figure 12c,f,i. The pore structure of the aerogel in FRABs after heat treatment at 400 °C is similar to that of the untreated one. However, the pore size after heat treatment at 600 °C decreased slightly. After heat treatment at 600 °C, the aerogel particles coalesced to form larger particles. Meanwhile, the size of the pores surrounded by these particles was reduced. This was consistent with the aerogel changes after heat treatment. The state of the individual fibers was also observed in Figure 12b,e,h. The adhesion of the aerogel to the fiber and evidence of the sintering of aerogel particles can be found after heat treatment at a temperature of 600 °C, as shown in Figure 12h. Woignier et al. studied the sintering behavior of silica aerogel and found that sintering due to a diffusional process occurs in the temperature range of 500–700 °C [43]. The sintering behavior of aerogel was influenced by the contact between fiber and aerogel, which had an effect on the thermal insulation properties of FRABs at high temperatures. Therefore, it is important to test the thermal insulation properties of FRABs at high temperatures.

### 3.4. The Performance of FRABs at High Temperature

The RT thermal conductivity of the FRABs after heat treatment cannot truly reflect the thermal insulation performance of the FRABs at high temperatures. The data tested in the actual service temperature are more convincing. Therefore, a heating table was used to simulate the performance of FRABs and glass wool at high temperatures, as presented in Section 2.3.3. The value of the surface temperature was recorded at different operating temperatures for samples with a thickness of 5 cm. The variation in surface temperature with heating time is shown in Figure 13. The cold surface temperature reached a relatively stable state after 120 min of heating at 200 °C and 300 °C, respectively. The time required for the cold surface temperature to reach a steady state is 300 min, at which point, the system reached a steady state at 600 °C. This time is much longer than 2 h, which was reported by Huafei Cai [26]. In this experiment, 5 cm thick FRABs were used, and it took a long time for the heat flow to pass through the FRABs due to their excellent insulation capacity. As a result, the aerogel particles were affected and aggregated relatively slowly. Glass wool was also tested in the same way under the same conditions for comparison. The heating power of glass wool and FRABs were tested at different temperatures and sample thicknesses, and the results are shown in Figure 14.

It is obvious that the heating power increased with increasing temperature. The heating power gradually decreased to a constant value with the increasing thickness of the tested materials. When the thickness is large enough, it is considered that heat will not be lost through the tested materials. In this case, the heating power is equal to the heat loss power of the heating table. 

The thermal conductivity of a homogeneous material does not change with direction. In the steady state without an internal heat source, it can be obtained according to Fourier’s law:(1)$$\lambda =\frac{\emptyset }{A}\cdot \frac{d}{t_{2}-t_{1}}$$
where *d* is the thickness of the tested material, *A* is the area of the tested material, $t_{2}$ is the operating temperature, $t_{1}$ is the temperature on the surface, and ∅ is the heat flow, i.e., the heat loss of the FRABs or glass wool.

The heat loss power φ of the plate heater is regarded as a fixed value at the same working temperature and has the following relationship with the heating power *P*:(2)P=∅+φ

It is approximately considered that the temperature difference $t_{2}-t_{1}$ is equal to the average value Δt at the same working temperature.
(3)$$P=\lambda A\Delta t\cdot \frac{1}{d}+\varphi$$

There is a linear relationship between the heating power *P* and the reciprocal thickness 1/*d*. The fittings are shown in Figure 15.

The thermal conductivity of glass wool and FRABs at different temperatures can be calculated according to Figure 15. The thermal conductivity comparison of glass wool and FRABs at high temperatures is shown in Figure 16. The results are consistent with Miros’s research which was measured on a two-plate guarded hot plate apparatus [33].

The thermal conductivity of FRABs is lower than that of glass fiber in the temperature range of 200 °C to 600 °C as shown in Figure 16. The pore size of FRABs is smaller than that of glass wool due to the filling of aerogel powders. According to the Knudsen effect [44], the smaller the pore size, the weaker the convective heat transfer of the gas and the lower the thermal conductivity. There is also an interfacial thermal resistance in the composite material [30,34,45]. The addition of aerogel creates more interfaces in FRABs, resulting in a higher thermal resistance.

In the practical application of insulation materials, heat loss is one of the most important parameters of interest to the user. Using the heating table designed in this experiment, not only the thermal conductivity of the material but also the corresponding heat loss can be calculated. According to the data in Figure 13 and Equation (2), the heat loss of FRABs and glass wool at different temperatures can be obtained using the following equation.
(4)∅=P−φ
where ∅ is the heat loss of FRABs or glass wool. *P* is the heating power. φ is the heat loss power of the heating table. The heat loss of FRABs and glass wool at an operating temperature of 600 °C is shown in Table 1. Heat loss refers to the heat passing through per square meter of insulation material per hour at the working temperature, so its unit is kJ/(h·m^2^). Using FRABs instead of glass fiber saves 656 kJ per square meter per hour when the thickness of insulation is 5 cm, and 5056 kJ per square meter when the thickness of the insulation is 1 cm, as shown in Figure 17. The heat loss of FRABs is much lower than that of glass wool. Using FRABs instead of glass wool is a very effective way to save energy.

## 4. Conclusions

In this work, the performances of aerogels, glass wool, and FRABs after heat treatment in the temperature range from RT to 600 °C were investigated. A heating table was designed that could simulate the scene of the thermal insulation material in the actual working process. The thermal insulation properties of glass wool and FRABs were investigated considering the working environment. The main conclusions can be summarized as follows:(1)The thermal conductivity of aerogel first decreased and then increased with the increase in heat treatment temperature, due to the change in the microstructure. The hydrophobicity of the aerogel disappeared after heat treatment at 400 °C.(2)The RT thermal conductivity of the glass fiber was less affected by the heat treatment temperature under 600 °C. A scanning electron microscope observation showed that the microstructure of glass fibers in FRABs remained stable after heat treatment.(3)The effect of heat treatment on the properties of FRABs was similar to that of aerogels. The contact of the aerogel with the fibers appeared to affect the sintering behavior of the aerogel particles at high temperatures, which may affect the performance of FRAB when in use.(4)Heat losses of FRAB and glass fibers were calculated and compared in service by using the heating table. When the working temperature is 600 °C and the thickness of the insulation is 5 cm, using FRAB instead of glass fiber can save 656 kJ per square meter per hour. Additionally, when the thickness of the insulation is 1 cm, the energy saved per square meter per hour is 5056 kJ. Using FRAB instead of glass wool is a very effective way to save energy.

## Figures and Tables

**Figure 1 materials-16-04888-f001:**
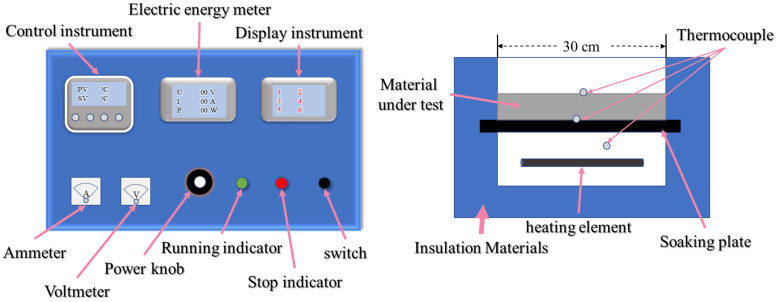
Schematic diagram of plate heater: console (**left**) and the heating device (**right**).

**Figure 2 materials-16-04888-f002:**
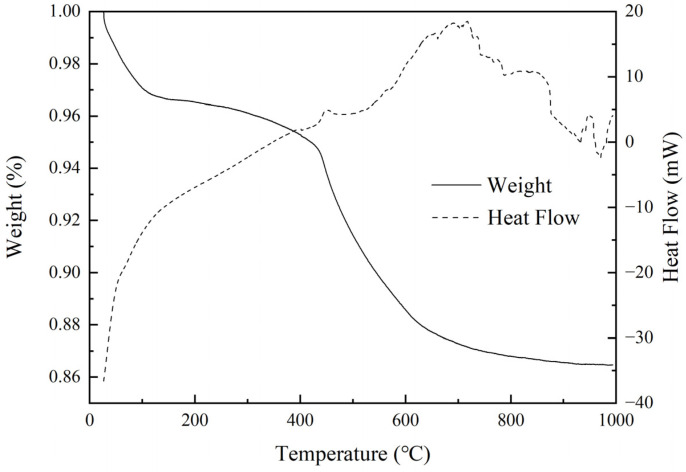
TG-DSC curves of the aerogel samples.

**Figure 3 materials-16-04888-f003:**
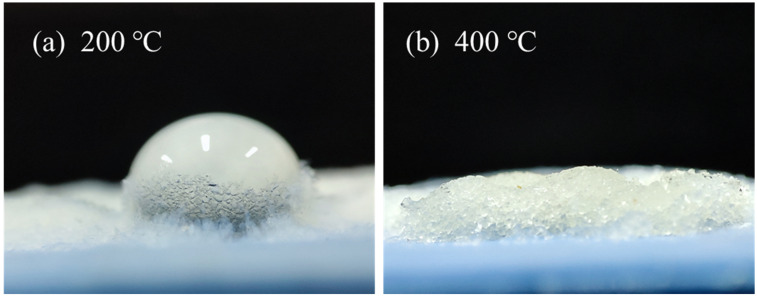
Droplet-spreading characteristics of aerogel after heat treatment at (**a**) 200 °C and (**b**) 400 °C.

**Figure 4 materials-16-04888-f004:**
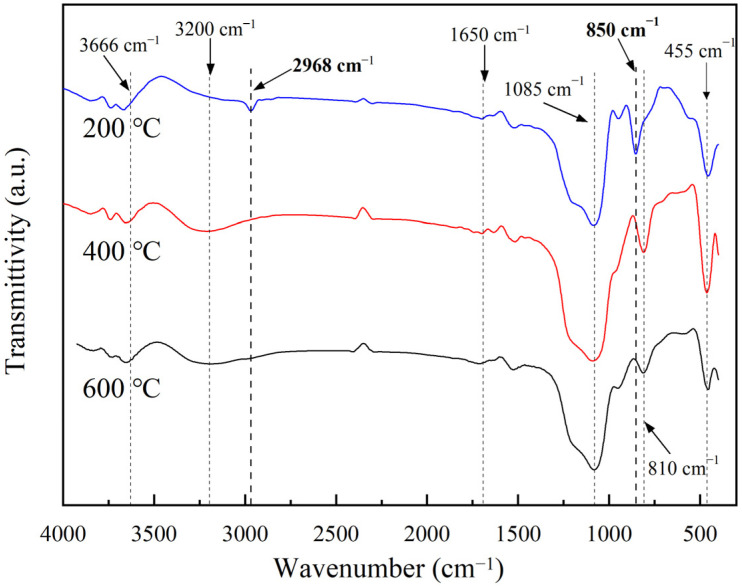
FT-IR spectra of the hydrophobic silica aerogel after heat treatment at 200 °C, 400 °C, and 600 °C.

**Figure 5 materials-16-04888-f005:**
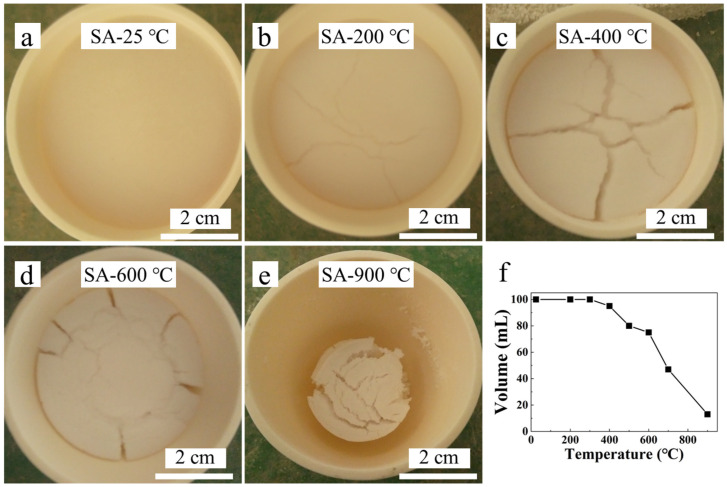
Photograph of the hydrophobic silica aerogel heat-treated at different temperatures (**a**) 25 °C, (**b**) 200 °C, (**c**) 400 °C, (**d**) 600 °C, (**e**) 900 °C, and (**f**) volume change.

**Figure 6 materials-16-04888-f006:**
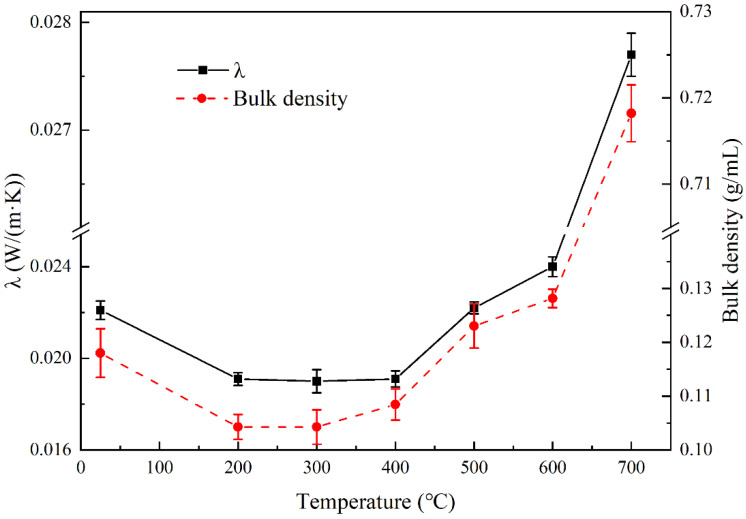
Bulk density and thermal conductivity of the silica aerogel samples after heat treatment from RT to 700 °C.

**Figure 7 materials-16-04888-f007:**
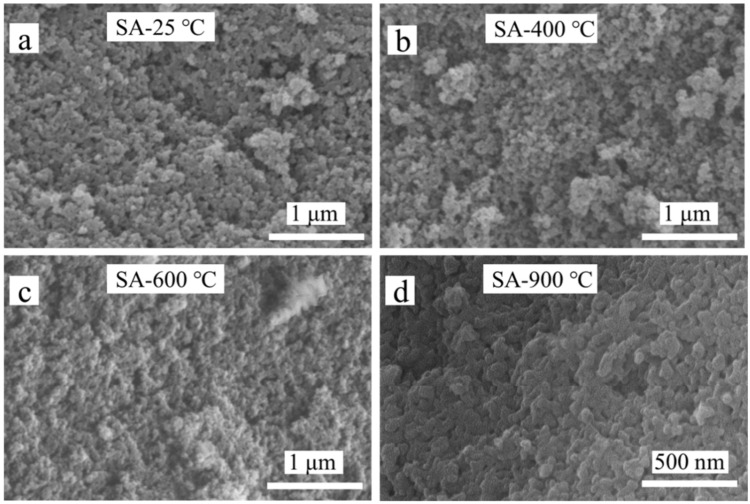
SEM images of silica aerogel after heat treatment at (**a**) 25 °C, (**b**) 400 °C, (**c**) 600 °C, (**d**) 900 °C.

**Figure 8 materials-16-04888-f008:**
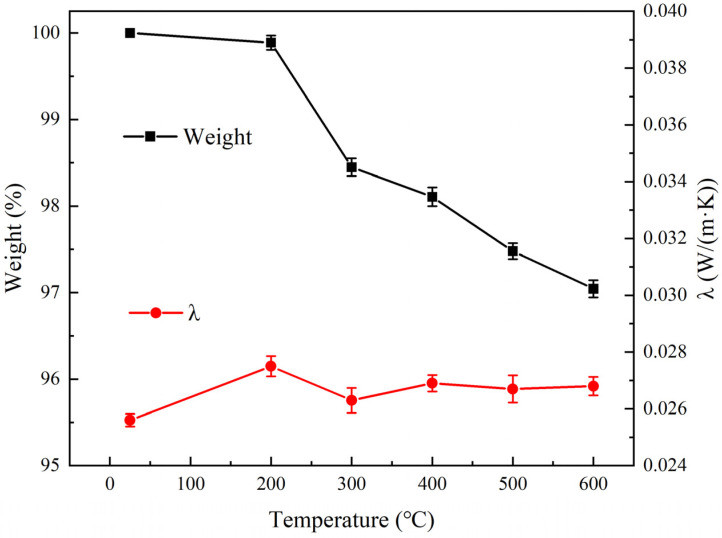
Weight loss and thermal conductivity of the glass wool after heat treatment from RT to 600 °C.

**Figure 9 materials-16-04888-f009:**
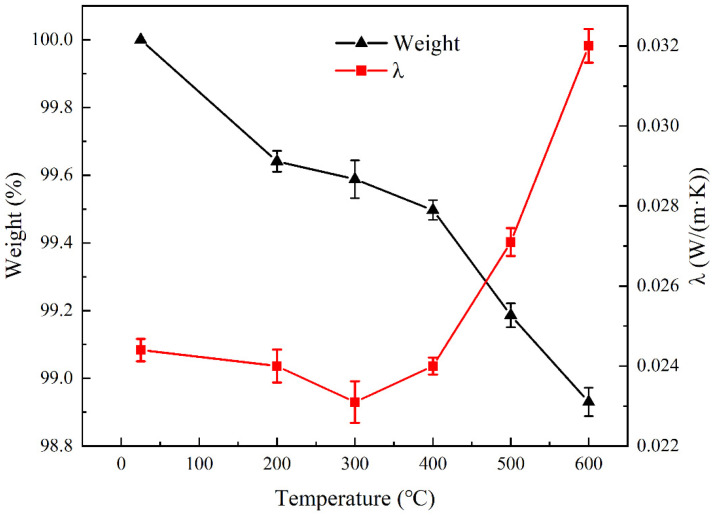
Weight loss and thermal conductivity of the FRAB after heat treatment from RT to 600 °C.

**Figure 10 materials-16-04888-f010:**
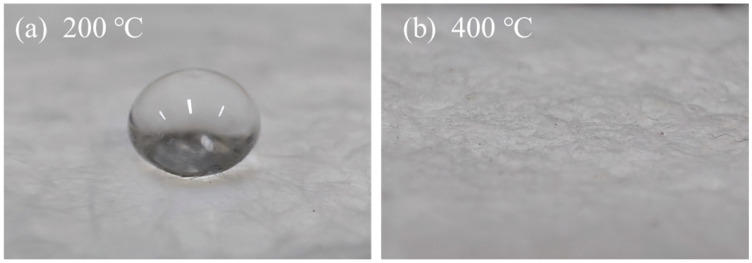
Droplet-spreading characteristics of FRAB after heat treatment at (**a**) 200 °C and (**b**) 400 °C.

**Figure 11 materials-16-04888-f011:**
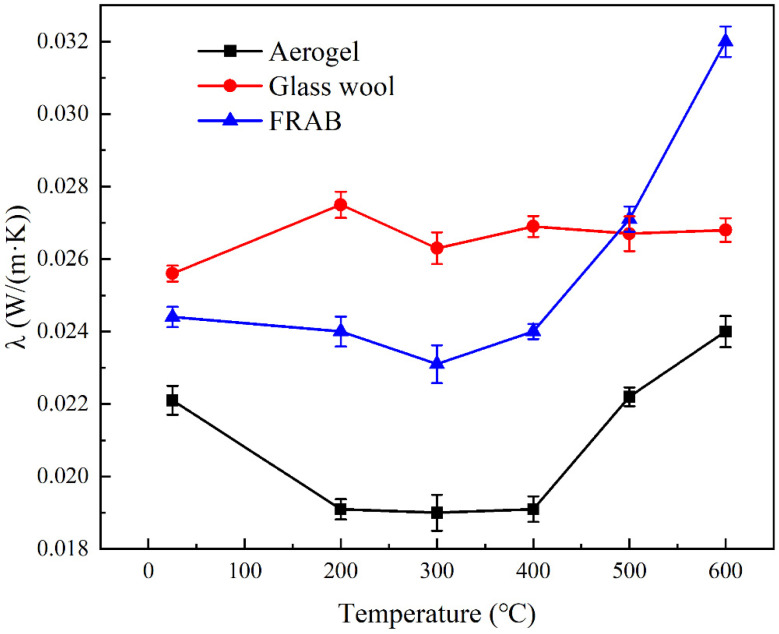
Thermal conductivity of the aerogel, glass wool, and FRAB after heat treatment from RT to 600 °C.

**Figure 12 materials-16-04888-f012:**
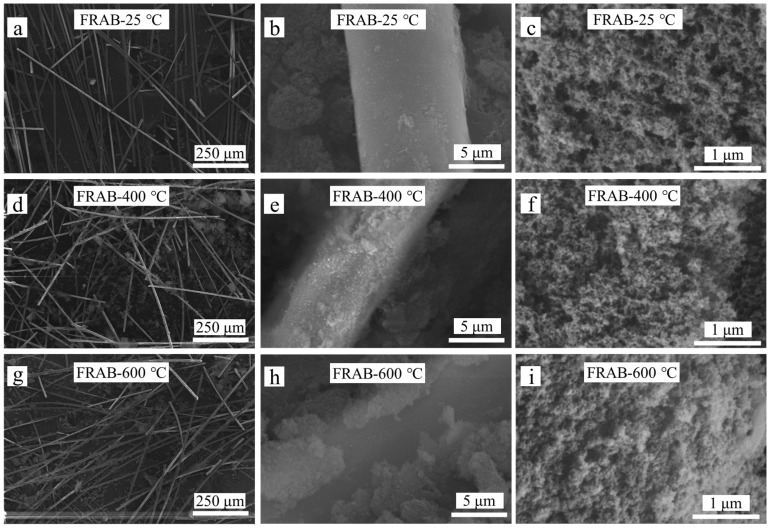
SEM images of FRABs before and after heat treatment. (**a**–**c**): untreated FRABs; (**d**–**f**): FRABs after heat treatment at 400 °C; (**g**–**i**): FRABs after heat treatment at 600 °C.

**Figure 13 materials-16-04888-f013:**
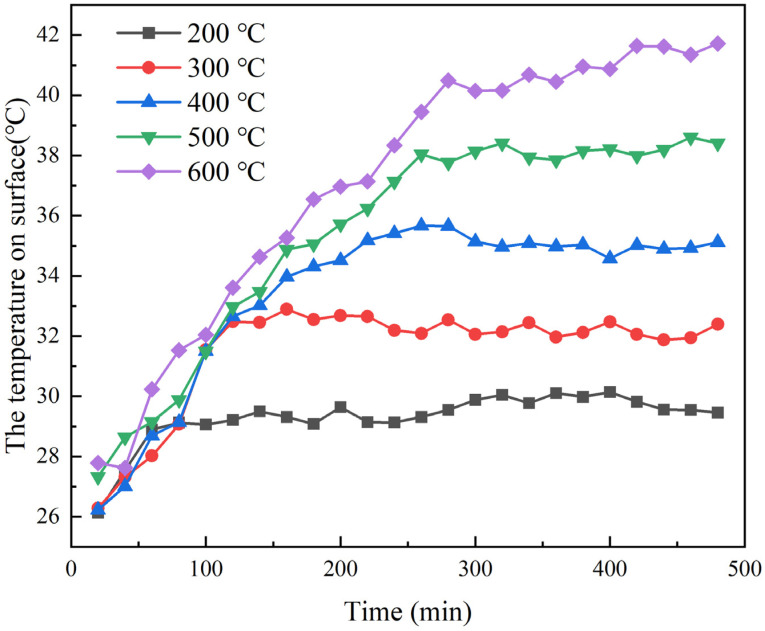
The temperature on surface changes with time at different temperatures.

**Figure 14 materials-16-04888-f014:**
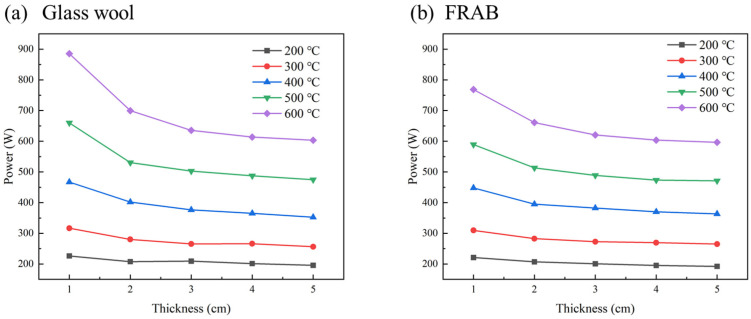
Heating power of (**a**) glass wool, and (**b**) FRABs changes with thickness at different temperatures.

**Figure 15 materials-16-04888-f015:**
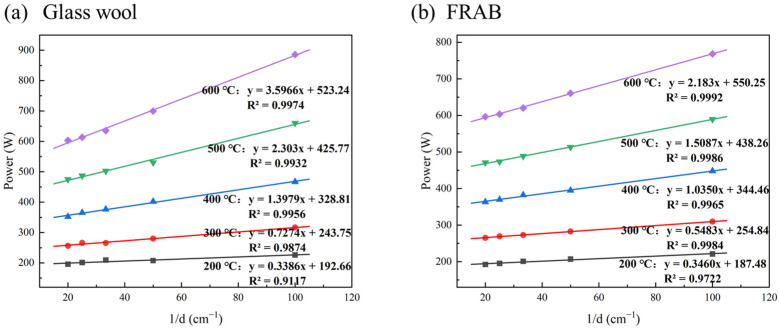
The relationship between heating power and the reciprocal thickness 1/*d*.

**Figure 16 materials-16-04888-f016:**
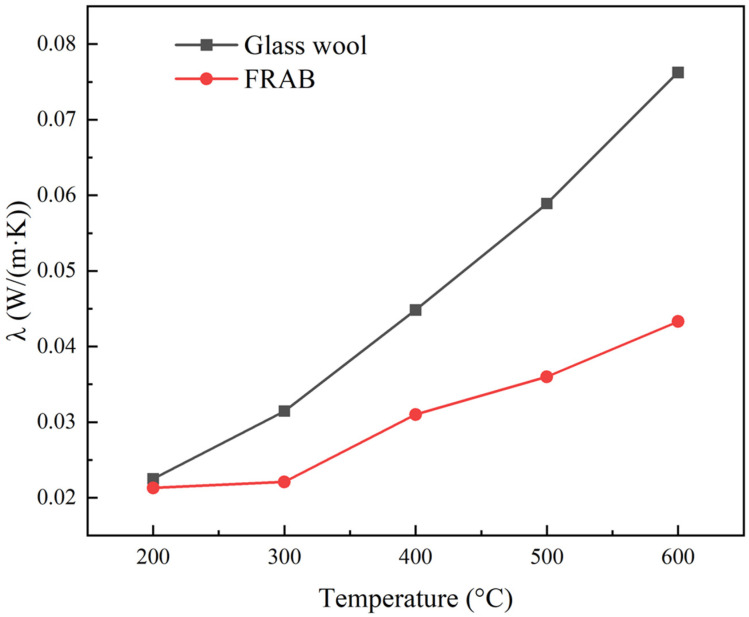
The thermal conductivity of glass wool and FRAB at high temperatures.

**Figure 17 materials-16-04888-f017:**
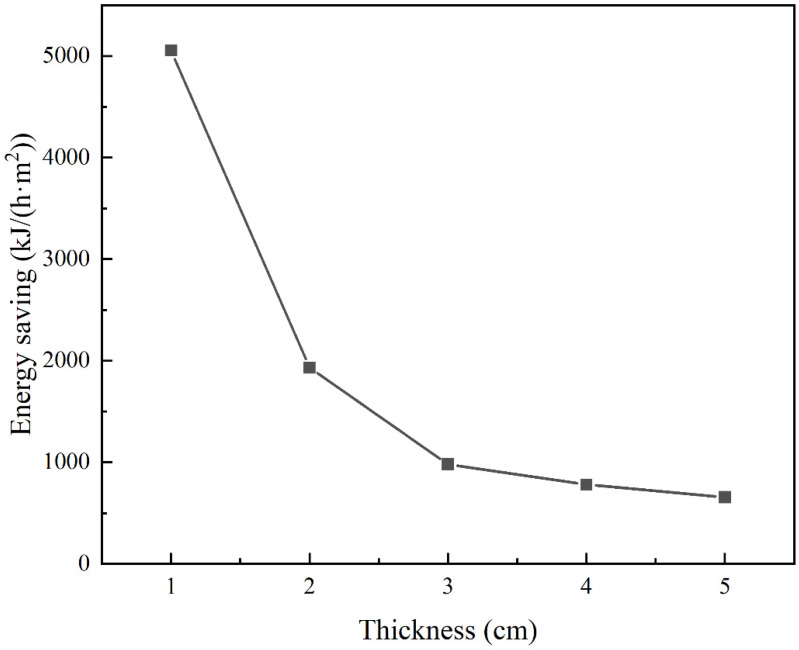
Energy savings change with thickness by using FRAB instead of glass fiber at 600 °C.

**Table 1 materials-16-04888-t001:** Heat loss of FRAB and glass fiber at different thicknesses at 600 °C.

Thickness (cm)	Heat Loss (kJ/(h·m^2^))	
	FRAB	Glass Fiber
1	9430.4	14,486.4
2	5126.4	7058.4
3	3506.4	4486.4
4	2834.4	3614.4
5	2542.4	3198.4

## Data Availability

The data that support the findings of this study are available on request from the corresponding author, Z.Z., upon reasonable request.

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
