# Peer review of "The Evolution of Insulation Performance of Fiber-Reinforced Silica Aerogel after High-Temperature Treatment"

_materials, 2023, doi:10.3390/ma16134888_

Round 1

Reviewer 1 Report

The problem posed in the study is of a quality that can attract the attention of the reader. The methods and techniques developed for the definition and solution of the problem are applicable. The results are scientifically verifiable.

1. What is the main question addressed by the research?

Heat treatment of fiber reinforced silica airgel at high temperatures at

different temperatures, and also what effect it has on the performance

of hydrophobic silica airgel powder and its base material glass wool

blanket are discussed. Although there are studies on these materials in

the literature, the effect of their behavior under heat treatment on

performance may contribute to the literature.

2. Do you consider the topic original or relevant in the field? Does it

address a specific gap in the field?

The subject includes methods and results close to similar studies. With

this aspect, its originality is weak, but it contains current topics

that may be of interest to the reader.

3. What does it add to the subject area compared with other published

material?

Fiber reinforced silica airgel used in the study and its material glass

wool and silica airgel are commercial products.

In addition, water glass was used in the preparation of silica airgel

using water glass as a precursor in the study.

With this feature, the material is commercially available and includes

standard features, making the test results reliable, but weakening the

depth and subjectivity of the study.

4. What specific improvements should the authors consider regarding the

methodology? What further controls should be considered?

The contents given in the figures must be improved. Units should be

written with explanation. Calculations must be taken into account in

modeling. Experimental methods were chosen appropriately.

5. Are the conclusions consistent with the evidence and arguments

presented and do they address the main question posed?

The results seem consistent and consistent with the problem. However,

careful examination of some results is recommended.

6. Are the references appropriate?

The references are sufficient in quantity and quality.

7. Please include any additional comments on the tables and figures.

Maximum attention should be paid to the content of the figures.

Author Response

Thank you very much for your nice work to review this manuscript. We really appreciate all your valuable comments and constructive suggestions on our manuscript entitled “The insulation performance evolution of fiber reinforced silica aerogel under high temperature”. (ID: materials-2461669). We have addressed the comments raised by the reviewers, and the amendments are highlighted in the revised manuscript. Detailed point by point responses to the reviewers’ comments are listed below.

Thanks again!

Reviewer: 1

Comment to the Author

The problem posed in the study is of a quality that can attract the attention of the reader. The methods and techniques developed for the definition and solution of the problem are applicable. The results are scientifically verifiable.

Response: Thank you very much for your positive comments on this manuscript. We have revised the manuscript carefully based on all reviewers’ comments. We hope the revised manuscript can meet the requirements of the journal now.

  1. What is the main question addressed by the research?

Heat treatment of fiber reinforced silica airgel at high temperatures at different temperatures, and also what effect it has on the performance of hydrophobic silica airgel powder and its base material glass wool blanket are discussed. Although there are studies on these materials in the literature, the effect of their behavior under heat treatment on performance may contribute to the literature.

Response: We appreciate your comment. The evolution of microstructure and thermal insulation performance of the fiber reinforced silica aerogel blanket (FRAB) after heat treatment at high temperature have been investigated in this work, which is important for industrial application.

  1. Do you consider the topic original or relevant in the field? Does it address a specific gap in the field?

The subject includes methods and results close to similar studies. With this aspect, its originality is weak, but it contains current topics that may be of interest to the reader.

Response: Thank you so much for your positive comment. The insulation performance of FRAB at service temperature is a topic of interest. In this work, the heat loss of FRAB during service at high temperature was measured by a homemade heating service, then energy saving efficiency was calculated compared with glass wool matrix. The obtained results can provide basic data and reference for practical applications of FRAB.

  1. What does it add to the subject area compared with other published material?

Fiber reinforced silica airgel used in the study and its material glass wool and silica airgel are commercial products. In addition, water glass was used in the preparation of silica airgel using water glass as a precursor in the study. With this feature, the material is commercially available and includes standard features, making the test results reliable, but weakening the depth and subjectivity of the study.

Response: We appreciate your comment. In this work, we use commercial materials with standard features as experimental materials to ensure the reliability of the results and therefor contribute to guiding the industrial applications of FRAB. We have done our best to revise the manuscript to strengthen the depth and subjectivity of this work.

  1. What specific improvements should the authors consider regarding the methodology? What further controls should be considered?

The contents given in the figures must be improved. Units should be written with explanation. Calculations must be taken into account in modeling. Experimental methods were chosen appropriately.

Response: Thank you very much for your comments and suggestions. Figures 1, 5, 8 and 9 have been modified and improved in the revised manuscript. The units of heat loss have been explained in the revised manuscript. We used the Fourier’law and simplified the calculation in order to make the calculation process more explicit.

  1. Are the conclusions consistent with the evidence and arguments presented and do they address the main question posed?

The results seem consistent and consistent with the problem. However, careful examination of some results is recommended.

Response: Thank you very much for your recognition and reminder. We have carefully calculated again and proofread the results, and some inappropriate points have been corrected in the revised manuscript.

  1. Are the references appropriate?

The references are sufficient in quantity and quality.

Response: Thank you very much for your recognition.

  1. Please include any additional comments on the tables and figures.

Maximum attention should be paid to the content of the figures.

Response: Thank you very much for the reminder. Inappropriate pictures have been modified.

Rui Gao

E-mail: gao_rui95@163.com

Name: Zhangjian Zhou

E-mail: zhouzhj@mater.ustb.edu.cn

Address: School of Materials Science and Engineering,

University of Science and Technology Beijing, Beijing 100083, China

Thank you again for your valuable comments and suggestions!

Reviewer 2 Report

The authors studied the effect of heat treatment at different temperatures on the performance of FRAB, hydrophobic silica aerogel powder and its base material glass The results showed that property evolution of the hydrophobic silica aerogel powder below 600 °C. The insulation performance of glass wool and FRAB at high temperature was studied by a heating table which was designed to simulate working conditions. The thermal conductivity of FRAB and glass wool at different temperatures was measured. The innovation of this paper is insufficient.

The reviewers have the following suggestions.

1. Manuscript writing needs polishing.

2. The author used fiber reinforced silica aerogel(FRAB), but its mechanical properties are not good enough.

3. Why only measure thermal insulation below 600 ° C?

4. What is the standard and basis of energy saving calculation? Are factors such as cost, quality, aperture and thickness considered in the calculation?

Manuscript writing needs polishing.

Author Response

Thank you very much for your nice work to review this manuscript. We really appreciate all your valuable comments and constructive suggestions on our manuscript entitled “The insulation performance evolution of fiber reinforced silica aerogel under high temperature”. (ID: materials-2461669). We have addressed the comments raised by the reviewers, and the amendments are highlighted in the revised manuscript. Detailed point by point responses to the reviewers’ comments are listed below.

Thanks again!

Reviewer: 2

Comment to the Author

The authors studied the effect of heat treatment at different temperatures on the performance of FRAB, hydrophobic silica aerogel powder and its base material glass. The results showed that property evolution of the hydrophobic silica aerogel powder below 600 °C. The insulation performance of glass wool and FRAB at high temperature was studied by a heating table which was designed to simulate working conditions. The thermal conductivity of FRAB and glass wool at different temperatures was measured. The innovation of this paper is insufficient. The reviewers have the following suggestions.

Response: Thank you very much for your comments and suggestions. It is true that many studies have been reported focused on the changes in the microstructure and properties of aerogels after heat treatment or at high temperatures. However, the thermal conductivity predicted by calculation cannot fully reflect the thermal insulation performance of the material under working conditions due to the complex high temperature environmental conditions. It is necessary to measure the thermal insulation performance of FRAB after heat treatment and compared with the matrix of glass wool. The experimental heat loss data obtained for high temperature systems using aerogels and their composites as insulation materials can provide more effective guidance for their practical application. This is the motivation for this work. We designed and made a heating table to simulate the actual working conditions, therefore, the heat loss of heat treated FRAB and glass wool at different temperatures and thicknesses can be measured and compared. These data combined with the evolution of the microstructure and thermal conductivity of aerogel and FRAB after heat treatment can help to better guide the application of aerogel-based insulation materials. The following are detailed responses to your suggestions one by one:

  1. Manuscript writing needs polishing.

Response: Thank you for your comments. We have carefully checked the typo errors and polishing the language of the manuscript. Then the manuscript was proofread by a native English speaker.

  1. The author used fiber reinforced silica aerogel (FRAB), but its mechanical properties are not good enough.

Response: Yes, we fully agree that the mechanical properties of FRAB are not good enough. Although some experimental works on Aerogel with good mechanical properties have been reported, aerogel usually needs to be made into composite materials with other materials (typically glass wool) for industrial application using as insulation materials due to high brittleness of commercial aerogel. Currently, the most typical commercial composite product is fiber reinforced silica aerogel board (FRAB). The tensile strength of the used commercial FRAB in this work is 200 kPa, which meets the ASTM C1728-23 requirements.

  1. Why only measure thermal insulation below 600 °C?

Response: Thank you for your comment. The selection of test temperature is based on the typical service temperature of FRAB, as the operating temperature of steam networks and most of high temperature equipment that requires insulation is typically at 300-600 °C, according to our investigation [1].

[1] Fedyukhin, A. V; Strogonov, K. V; Soloveva, O. V; Solovev, S.A.; Akhmetova, I.G.; Berardi, U.; Zaitsev, M.D.; Grigorev, D. V Aerogel Product Applications for High-Temperature Thermal Insulation. energies 2022, 15, 7792.

  1. What is the standard and basis of energy saving calculation? Are factors such as cost, quality, aperture and thickness considered in the calculation?

Response: Thank you very much for your comment. We measured the heat loss using FRAB and glass wool with the same thickness under the same environmental conditions. The difference in obtained results based on these two materials is considered as the energy savings, which refers to the reduction of energy loss per hour per square meter by replacing glass wool with FRAB of the same thickness. The influence of factors such as cost is indeed also important for the economic calculation. However, there have been frequent changes in raw materials costs recently, and the economic calculation is a bit beyond the scope of this work. We will consider these calculation work in the near future. Thank you again for your valuable comment.

Rui Gao

E-mail: gao_rui95@163.com

Name: Zhangjian Zhou

E-mail: zhouzhj@mater.ustb.edu.cn

Address: School of Materials Science and Engineering,

University of Science and Technology Beijing, Beijing 100083, China

Thank you again for your valuable comments and suggestions!

Reviewer 3 Report

The work is interesting. However in its current form, it cannot be published. I recommend to publish it after resolving the following concerns?

 Some comments

       i.          The background of the work is not enough.

     ii.          The mechanical strength investigation would be interesting here.

    iii.          The connection of one sentence to other are missing in many cases. Need to revise extensively.

   iv.          English need to be checked carefully by a native speaker.

     v.          Please rewrite the last paragraph of introduction section. This is not experimental section

   vi.          In the Materials and methods section. The title Materials is not suitable for the paragraph. Change it.

   vii.          anti-symmetric stretching/ antisymmetric stretching please write in unique style. What do you mean by antisymmetric stretching? Did you mean asymmetric stretching?

viii.          Page 1:  FRAB were applied in building and show very good performance. Please cite some of the good performance

    ix.          delete FTIR, SEM from the title 2.3.2. Microstructural Characterization (FTIR, SEM)

     x.          After 600 °C, the weight loss of aerogel slows down. Why?

   xi.          Increase the font size for figure 5f and use scale bar for rest of the images 5a-5e.

  xii.          In page 7- ….the volume shrinkage of the aerogel is very serious,…. What do you mean?

   The connection of one sentence to other are missing in many cases. Need to revise extensively.

There are many grammatical issues

Author Response

Thank you very much for your nice work to review this manuscript. We really appreciate all your valuable comments and constructive suggestions on our manuscript entitled “The insulation performance evolution of fiber reinforced silica aerogel under high temperature”. (ID: materials-2461669). We have addressed the comments raised by the reviewers, and the amendments are highlighted in the revised manuscript. Detailed point by point responses to the reviewers’ comments are listed below.

Thanks again!

Reviewer: 3

Comment to the Author

The work is interesting. However in its current form, it cannot be published. I recommend to publish it after resolving the following concerns?

Response: Thank you very much for your comments on this manuscript. The comments you given are exactly the key issues of this manuscript. We have carefully revised and improved the manuscript accordingly. The modifications based on your comments make the revised manuscript more logical. The followings are responses to your comments or questions one by one:

1. The background of the work is not enough.

Response: Thank you for your suggestion. The changes in microstructure and insulation performance of FRAB at service temperature is a topic of interest. In this work, the heat loss of FRAB during service at high temperature was measured by a homemade heating service, then energy saving efficiency was calculated compared with glass wool matrix. We use commercial materials with standard features as experimental materials to ensure the reliability of the results. Therefor the obtained results can provide basic data and reference for practical applications of FRAB. The related information have been added in the part of Introduction to strengthen the background of this work in the revised manuscript.

2. The mechanical strength investigation would be interesting here.

Response: Thank you very much for your suggestion. The change in mechanical strength of aerogel after heat treatment is also interesting and worth investigation. The tensile strength of the used commercial FRAB in this work is 200 kPa, which meets the ASTM C1728-23 requirements. Currently commercially available Aerogel are very brittle. Therefore, aerogel usually needs to be made into composite materials with other materials (typically glass wool) when using as insulation materials for industrial application. In this case, mechanical strength of FRAB is usually very low, while thermal conductivity and energy saving are considered as most important issues.

3. The connection of one sentence to other are missing in many cases. Need to revise extensively.

Response: Thank you for your suggestion. The manuscript has been extensively revised to make it more smooth, concise and logical.

4. English need to be checked carefully by a native speaker.

Response: The authors apologize for the poor language of our manuscript. We have carefully checked the typo errors and polishing the language of the manuscript. Then the manuscript was proofread by a native English speaker. We hope that the language in the revised manuscript have been substantially improved.

5. Please rewrite the last paragraph of introduction section. This is not experimental section.

Response: Thank you very much for your suggestion. The last paragraph of Introduction section has been rewritten in the revised manuscript.

6. In the Materials and methods section. The title Materials is not suitable for the paragraph. Change it.

Response: Thank you for your comment. The title Materials has been modified as Experimental material.

7. anti-symmetric stretching/ antisymmetric stretching please write in unique style. What do you mean by antisymmetric stretching? Did you mean asymmetric stretching?

Response: Yes, asymmetric stretching is the correct expression, which have been revised in the revised manuscript. Sorry for this mistake.

8. Page 1: FRAB were applied in building and show very good performance. Please cite some of the good performance.

Response: Thank you for your suggestion. FRAB were applied in building and show very good performance including very low thermal conductivity (about 0.0248 W/(m·K) at room temperature), light weight, very good sound isolation, and excellent waterproof performance. Relevant content has been added in Page 1 in the revised manuscript.

9. delete FTIR, SEM from the title 2.3.2. Microstructural Characterization (FTIR, SEM)

Response: Thank you very much for your suggestion, the relevant content has been modified in the revised manuscript.

10. After 600 °C, the weight loss of aerogel slows down. Why?

Response: Thank you for your comment. The weight loss of aerogel during heating process is mainly due to the decomposition of hydrophobic agent (methyl group) which is basically completed before 600 °C. Therefore, the weight loss of FRAB slows down after 600 °C,

11. Increase the font size for figure 5f and use scale bar for rest of the images 5a-5e.

Response: Thank you very much for your suggestion, the pictures in the manuscript have been revised based on your comments.

12. In page 7- ….the volume shrinkage of the aerogel is very serious,…. What do you mean?

Response: Sorry for the unclear expression. According to Fig. 5e and f, the volume of the aerogel decreases substantially after heat treatment, which indicate obvious volume shrinkage of the aerogel after high temperature treatment. We have revised the expression.

Rui Gao

E-mail: gao_rui95@163.com

Name: Zhangjian Zhou

E-mail: zhouzhj@mater.ustb.edu.cn

Address: School of Materials Science and Engineering,

University of Science and Technology Beijing, Beijing 100083, China

Thank you again for your valuable comments and suggestions!

Reviewer 4 Report

Someone needs to rerite the paper to improve the English usage. There are other issues to address as well. Abstract, groups were .... were calculated. P2, process of areogels. P2, there are way too many significant figures in many of those data. Figure 1 needs a scale. Figure 4, identify all peaks. Figure 5 give a scale. Figure 8 in black give error bars. Figure 9 in black give error bars. Figures 13, 14, give error bars. Figure 16, 17, give error bars.

1. What is the main question addressed by the research? The main question involves evaluation of fiber reinforced silica aerogels as insulating agents.

2. Do you consider the topic original or relevant in the field? Does it address a specific gap in the field? I think the work is original and addresses a gap.

3. What does it add to the subject area compared with other published material? There are new data along the lines of silica aerogels.

4. What specific improvements should the authors consider regarding the methodology? What further controls should be considered? Methodology and controls are OK.

5. Are the conclusions consistent with the evidence and arguments presented and do they address the main question posed? Yes.

6. Are the references appropriate? Yes.

7. Please include any additional comments on the tables and figures. They are OK.

The paper must be rewritten. 

Author Response

Thank you very much for your nice work to review this manuscript. We really appreciate all your valuable comments and constructive suggestions on our manuscript entitled “The insulation performance evolution of fiber reinforced silica aerogel under high temperature”. (ID: materials-2461669). We have addressed the comments raised by the reviewers, and the amendments are highlighted in the revised manuscript. Detailed point by point responses to the reviewers’ comments are listed below.

Thanks again!

Reviewer: 4

Comment to the Author

Someone needs to rerite the paper to improve the English usage. There are other issues to address as well. Abstract, groups were .... were calculated. P2, process of areogels. P2, there are way too many significant figures in many of those data.

Response: Thanks for your careful reading. We have done our best to revise and improve the English usage of the manuscript. Now we have carefully worked on checking the typo errors of the manuscript. Grammar errors and expression problems have been carefully checked and revised.

Figure 4, identify all peaks.

Response: Thank you for your suggestion. All peaks were identified in Figure 4, and were explained in the revised manuscript.

Figure 1 needs a scale. Figure 5 give a scale. Figure 8 in black give error bars. Figure 9 in black give error bars. Figures 13, 14, give error bars. Figure 16, 17, give error bars.

Response: Thank you very much for your comments and suggestions. Scales were added to Figures 1 and 5. Error bars are calculated and added in Figure 8 and Figure 9. Figures 1, 4, 5, 8 and 9 have been appropriately modified in the revised manuscript. As for Fig. 13 and 14, the surface temperature of the tested samples and the Power are directly readings from the heating table, therefore no error bars. Similarly for Fig 16 and 17, which are obtained by calculation according to data in Fig. 13 and 14.

Rui Gao

E-mail: gao_rui95@163.com

Name: Zhangjian Zhou

E-mail: zhouzhj@mater.ustb.edu.cn

Address: School of Materials Science and Engineering,

University of Science and Technology Beijing, Beijing 100083, China

Thank you again for your valuable comments and suggestions!

Round 2

Reviewer 2 Report

Silica aerogels have been widely used in aerospace and other fields, but reviewers could not see the characteristics of the research from the revised manuscript content, and the innovation was insufficient. The author did not give a definite innovation point in the answer to the question.

Author Response

Thank you very much for your nice work to review this manuscript. We really appreciate all your valuable comments and constructive suggestions on our manuscript entitled “The insulation performance evolution of fiber reinforced silica aerogel under high temperature”. (ID: materials-2461669). We have addressed the comments raised by the reviewers, and the amendments are highlighted in the revised manuscript. Detailed point by point responses to the reviewers’ comments are listed below.

Thanks again!

Reviewer: 2

Comment to the Author

Silica aerogels have been widely used in aerospace and other fields, but reviewers could not see the characteristics of the research from the revised manuscript content, and the innovation was insufficient. The author did not give a definite innovation point in the answer to the question.

Response: Thank you very much for your comments. A number of works have been performed on characterization of silica aerogels. We fully agree that silica aerogels have been widely investigated and applied in aerospace and other fields. As pure aerogel thermal insulation material, glass window combining silica aerogel has also been reported as an engineering application [1][2]. But for applications in high-temperature industry, such as thermal insulation for high-temperature steam pipes, only aerogel composites, mainly FRAB, instead of pure aerogel are commercially applied. The evaluation of their performance is extremely important. The recent research on new aerogel insulation materials are mainly focused on the preparation and microstructure investigation, only the thermal conductivity has been used as a single indicator to evaluate the thermal insulation performance, and the engineering application performance evaluation is always absent. The results of pure aerogel obtained in the laboratory cannot be used to accurately evaluate the service behavior of composite materials like FRAB. In order to better understand the service behavior of FRAB, it is necessary to compare and evaluate aerogel, the matrix of glass wool, and FRAB respectively, under the same simulated actual operating conditions. This is the motivation of this work. A heating table has been designed and constructed to simulate actual operating conditions, therefore, the heat loss of heat treated FRAB, as well as glass wool at different temperatures and thicknesses can be measured and compared. These data combined with the evolution of the microstructure and thermal conductivity of aerogel and FRAB after heat treatment can help to better guide the application of aerogel-based insulation materials.

The part of introduction has been revised in the manuscript to strengthen the innovation of this manuscript. We hope it can meet the requirement now.

[1] Liu, S.; Wu, X.; Li, Y.; Cui, S.; Shen, X.; Tan, G. Hydrophobic In-Situ SiO2-TiO2 Composite Aerogel for Heavy Oil Thermal Recovery: Synthesis and High Temperature Performance. Appl. Therm. Eng. 2021, 190, 116745, doi:10.1016/j.applthermaleng.2021.116745.

[2] Li, D.; Zhang, C.; Li, Q.; Liu, C.; Arıcı, M.; Wu, Y. Thermal Performance Evaluation of Glass Window Combining Silica Aerogels and Phase Change Materials for Cold Climate of China. Appl. Therm. Eng. 2020, 165, 114547, doi:10.1016/j.applthermaleng.2019.114547.

Rui Gao

E-mail: gao_rui95@163.com

Name: Zhangjian Zhou

E-mail: zhouzhj@mater.ustb.edu.cn

Address: School of Materials Science and Engineering,

University of Science and Technology Beijing, Beijing 100083, China

Thank you again for your valuable comments and suggestions!

Reviewer 3 Report

Accept in present form

Author Response

Thank you very much for your nice work to review this manuscript. We really appreciate all your valuable comments and constructive suggestions on our manuscript entitled “The insulation performance evolution of fiber reinforced silica aerogel under high temperature”. (ID: materials-2461669). 

Thanks again!

Rui Gao

E-mail: gao_rui95@163.com

Name: Zhangjian Zhou

E-mail: zhouzhj@mater.ustb.edu.cn

Address: School of Materials Science and Engineering,

University of Science and Technology Beijing, Beijing 100083, China

Thank you again for your valuable comments and suggestions!

Reviewer 4 Report

Changes have been made and the paper is acceptable. 

Author Response

(The authors gave the same response as above.)
